# The Novel SSTR3 Agonist ITF2984 Exerts Antimitotic and Proapoptotic Effects in Human Non-Functioning Pituitary Neuroendocrine Tumor (NF-PitNET) Cells

**DOI:** 10.3390/ijms25073606

**Published:** 2024-03-23

**Authors:** Genesio Di Muro, Rosa Catalano, Donatella Treppiedi, Anna Maria Barbieri, Federica Mangili, Giusy Marra, Sonia Di Bari, Emanuela Esposito, Emma Nozza, Andrea G. Lania, Emanuele Ferrante, Marco Locatelli, Daniela Modena, Christian Steinkuhler, Erika Peverelli, Giovanna Mantovani

**Affiliations:** 1Department of Clinical Sciences and Community Health, University of Milan, 20122 Milan, Italy; genesio.dimuro@uniroma1.it (G.D.M.); rosa.catalano@unimi.it (R.C.); anna.barbieri@unimi.it (A.M.B.); giusy.marra@unimi.it (G.M.); sonia.dibari@unimi.it (S.D.B.); emanuela.esposito1@unimi.it (E.E.); emma.nozza@unimi.it (E.N.); 2Department of Experimental Medicine, University Sapienza of Rome, 00100 Rome, Italy; 3Endocrinology Unit, Fondazione Istituto di Ricovero e Cura a Carattere Scientifico (IRCCS) Ca’ Granda Ospedale Maggiore Policlinico, 20122 Milan, Italy; donatella.treppiedi@policlinico.mi.it (D.T.); federica.mangili@policlinico.mi.it (F.M.); emanuele.ferrante@policlinico.mi.it (E.F.); 4PhD Program in Experimental Medicine, University of Milan, 20100 Milan, Italy; 5Department of Biomedical Sciences, Humanitas University, 20090 Pieve Emanuele, Italy; andrea.lania@hunimed.eu; 6Endocrinology and Diabetology Unit, Istituto di Ricovero e Cura a Carattere Scientifico (IRCCS) Humanitas Research Hospital, 20089 Rozzano, Italy; 7Neurosurgery Unit, Fondazione Istituto di Ricovero e Cura a Carattere Scientifico (IRCCS) Ca’ Granda Ospedale Maggiore Policlinico, 20122 Milan, Italy; marco.locatelli@unimi.it; 8Department of Pathophysiology and Transplantation, University of Milan, 20122 Milan, Italy; 9Preclinical R&D, Italfarmaco Group, Cinisello Balsamo, 20092 Milan, Italy; d.modena@italfarmacogroup.com (D.M.);

**Keywords:** NF-PitNETs, SSTRs, ITF2984

## Abstract

Somatostatin receptor ligands (SRLs) with high affinity for somatostatin receptors 2 and 5 (SSTR2 and SSTR5) are poorly efficacious in NF-PitNETs, expressing high levels of SSTR3. ITF2984 is a pan-SSTR ligand with high affinity for SSTR3, able to induce SSTR3 activation and to exert antitumoral activity in the MENX rat model. The aim of this study was to test ITF2984’s antiproliferative and proapoptotic effects in NF-PitNET primary cultured cells derived from surgically removed human tumors and to characterize their SSTR expression profile. We treated cells derived from 23 NF-PitNETs with ITF2984, and a subset of them with octreotide, pasireotide (SRLs with high affinity for SSTR2 or 5, respectively), or cabergoline (DRD2 agonist) and we measured cell proliferation and apoptosis. SSTR3, SSTR2, and SSTR5 expression in tumor tissues was analyzed by qRT-PCR and Western blot. We demonstrated that ITF2984 reduced cell proliferation (−40.8 (17.08)%, *p* < 0.001 vs. basal, *n* = 19 NF-PitNETs) and increased cell apoptosis (+41.4 (22.1)%, *p* < 0.001 vs. basal, *n* = 17 NF-PitNETs) in all tumors tested, whereas the other drugs were only effective in some tumors. In our model, SSTR3 expression levels did not correlate with ITF2984 antiproliferative nor proapoptotic effects. In conclusion, our data support a possible use of ITF2984 in the pharmacological treatment of NF-PitNET.

## 1. Introduction

Non-functioning pituitary neuroendocrine tumors (NF-PitNETs) are generally benign tumors that account for more than one-third of all PitNETs. They are not associated with clinical evidence of hormonal hypersecretion, but patients present neurological symptoms such as headache and visual field defects due to the mass effect and clinical signs of pituitary hormone deficiencies [1,2,3]. The first-line treatment is transsphenoidal surgery, but recurrences are frequent and additional treatment is needed in the vast majority of patients [4,5,6]. However, no effective medical therapy is available for NF-PitNETs.

In contrast, for hormone-secreting PitNETs, drugs targeting somatostatin or dopamine receptors have been approved. Indeed, long-acting somatostatin receptor ligands (SRLs) are currently used in the treatment of growth hormone (GH)- and adrenocorticotropic hormone (ACTH)-secreting PitNETs, and agonists of dopamine receptor type 2 (DRD2) represent the first-choice treatment for patients with prolactin (PRL)-secreting PitNETs, although a subgroup of patients is resistant to these drugs [7,8,9]. 

The therapeutic action of SRLs is mediated by the activation of somatostatin receptors (SSTRs), a family of G-protein-coupled receptors (GPCRs) that include five different subtypes [10]. Upon agonist challenge, SSTRs trigger a complex signaling cascade starting from the interaction of activated receptors with specific G-proteins and involving a large number of different downstream proteins [10]. The coupling of SSTRs with inhibitory G proteins represents the basis of the modulation of several enzyme activities, such as adenylyl cyclase and phosphotyrosine phosphatases (PTPs), that, in turn, control the activity of downstream signaling molecules such as ERK1/2, and of intracellular levels of potassium and calcium ions. While all SSTR subtypes are able to mediate cell proliferation inhibition, only SSTR2 and SSTR3 have been associated with proapoptotic effects [11,12,13]. 

The first-generation SRLs, octreotide and lanreotide, have a high binding affinity for SSTR2 and, to a lesser extent, for SSTR5 and SSTR3, whereas the second-generation SRL pasireotide is a multireceptor ligand that binds with higher affinity to SSTR5 (40-fold), SSTR1 (30-fold), and SSTR3 (5-fold) and with the same affinity to SSTR2 when compared with octreotide [14]. 

Drugs that target DRD2, such as cabergoline, another GPCR coupled to inhibitory G proteins, are able to induce a reduction in cell proliferation and hormone secretion in PitNETs.

Studies testing receptor expression profiles have demonstrated that NF-PitNETs mainly express SSTR3, DRD2, and, to a lesser degree, SSTR2 and SSTR5 [15,16,17,18]. However, NF-PitNETs are generally unresponsive to SRLs targeting SSTR2 and SSTR5, and controversial data are reported for DRD2 agonists [19,20,21]. 

Interestingly, a novel SSTR3 agonist, ITF2984, has been recently characterized [22]. Although this molecule was originally developed as a pan-SSTR agonist, it shows a 10-fold improved affinity for SSTR3 compared to octreotide or pasireotide; it acts as a full agonist of SSTR3 in in vitro assays, promoting SSTR3 internalization and phosphorylation, and inducing G-protein signaling [22]. Moreover, ITF2984 has been tested in vivo in the MENX rat model, characterized by the spontaneous development of pituitary tumors that share features with human gonadotroph tumors [23,24]. ITF2984 demonstrated significant antitumor activity in female MENX rats, who showed higher expression levels of SSTR3 compared to males [22]. However, the effects of ITF2984 on cell growth have never been tested in primary cultured human NF-PitNET cells, nor in other cell lines. 

The aim of the present study was to assess the antiproliferative and proapoptotic efficacy of ITF2984 in vitro in primary cultured cells derived from surgically removed human NF-PitNETs. The effects of this drug were then correlated with SSTR expression, the lineage derivation of the tumors, and the sex of the patients.

## 2. Results

### 2.1. ITF2984 Effects on Cell Proliferation in Primary Cultured NF-PitNET Cells

To investigate the antiproliferative effects of ITF2984 in NF-PitNETs, we treated dispersed primary cultured cells derived from 19 surgically removed human NF-PitNETs with different concentrations of the compound for 72 h. We found that ITF2984 at the concentration of 100 nM reduced cell proliferation in all tumors, with a range of reduction from −21.6% to −64.4% (median reduction −40.8 (17.08) %, *p* < 0.001 vs. basal, *n* = 19 NF-PitNETs) (Figure 1a). A subset of tumors was also treated with pasireotide, octreotide, or cabergoline at 100 nM. Considering a reduction in cell proliferation of at least 20%, two out of six tumors were responsive to pasireotide, one out of five to octreotide, and one out of eight to cabergoline. Interestingly, one tumor was responsive to all drugs. As shown in Figure 1b, only ITF2984 reduced cell median proliferation, while pasireotide, octreotide, and cabergoline did not affect median cell proliferation.

In order to establish the lineage derivation of the tested tumors, the expression of the marker of gonadotrophic lineage steroidogenic factor-1 (SF1) was evaluated by RT-qPCR analysis. Our results showed a positive expression of *SF1* in 15/19 tumors. Among the four samples negative for *SF1*, one was positive for *PIT1*. No difference was found between the group of *SF1*-positive or -negative tumors in terms of the efficacy of ITF2984 in reducing cell proliferation (Appendix A).

### 2.2. ITF2984 Effects on Cell Apoptosis in Primary Cultured NF-PitNET Cells

To test the proapoptotic effects of ITF2984, we measured caspase 3/7 activity in cells incubated for 48 h with increasing concentrations of this drug. Our results obtained in 17 NF-PitNETs showed a statistically significant increase in cell apoptosis after incubation with ITF2984 at 10 nM (+25.6 (13.5)%, *p* < 0.01) and 100 nM (+41.4 (22.1)%, *p* < 0.001) (Figure 2a). The increase was at least +20.8% with a maximum increase of +93.4%. A subset of tumors was also treated with pasireotide, octreotide, or cabergoline, and we found 2/7, 1/7, and 2/10 tumors, respectively, in which the treatment induced an increase of at least 20% of the apoptosis. Considering the median of all tumors treated, no apoptosis induction was observed after treatment with pasireotide, octreotide, or cabergoline (Figure 2b). No difference was found in the ability of ITF2984 in promoting cell apoptosis in the *SF1*-positive (*n* = 15) or -negative (*n* = 2) samples (Appendix A).

### 2.3. Analysis of SSTR3, SSTR2, and SSTR5 Expression

The SSTR3 expression level in tumor tissues was evaluated by both RT-qPCR and Western blot analyses (Figure 3a,b). Our data showed that the SSTR3 transcript was expressed in all NF-PitNETs analyzed (*n* = 21) except one (#7). The median *SSTR3* expression level was comparable to that of *SSTR2* and *SSTR5*, whereas *SSTR2* was significantly higher than *SSTR5* (Figure 3a). No correlation was found between *SSTR3* and *SSTR2* or *SSTR3* and *SSTR5*, while we found a positive correlation between the *SSTR2* and *SSTR5* transcript levels (ρ = 0.76, *p* < 0.001) (Appendix A). 

Proteins were available for 17 tumors. Western blot analysis revealed SSTR3 protein expression in 16 samples, and SSTR2 and SSTR5 in all samples (Figure 3b). A low but detectable expression of SSTR3 was found in sample #7 that was negative for SSTR3 at RT-qPCR analysis. 

A positive correlation was found between SSTR3 and SSTR5 protein levels (ρ = 0.79, *p* < 0.001), but not between those of SSTR3 and SSTR2 (Figure 3c). 

Moreover, SSTR3 expression, both at the transcript and protein level, did not correlate with the inhibition of cell proliferation nor with apoptosis increase after ITF2984 treatment. Similarly, no correlation was found considering SSTR2 or SSTR5 expression levels (Appendix A). 

No difference in SSTR3 mRNA or protein expression between males and females was found (Appendix A) and, similarly, no difference in ITF2984 efficacy in males and females was detected (Appendix A). Considering *SF1* positive or negative tumors, no difference in SSTR3 expression was found (Appendix A).

## 3. Discussion

The present work first investigated the effects of the pan-SSTR agonist with high affinity for SSTR3, ITF2984, on the growth of human primary cultured cells derived from NF-PitNETs. 

No data regarding ITF2984’s effects on cell proliferation are available in the literature. 

Modena and coauthors tested ITF2984 in vivo in the MENX rat model. These Sprague Dawley rats bear germline frameshift mutations in the *p*27 gene and are characterized by the spontaneous development of a multiple endocrine syndrome within their first year of life with 100% penetrance, with pituitary tumors that share features with human gonadotroph tumors [23,24]. ITF2984 demonstrated significant antitumor activity in the female rats, who showed higher expression levels of SSTR3 compared to the males [22].

In agreement, our data demonstrated the significant efficacy of ITF2984 in reducing cell proliferation and in increasing cell apoptosis in human primary cultured NF-PitNET cells, with a maximal effect at the dose of 100 nM. A decrease in SRL’s effect with increasing dose has been previously observed [25], possibly due to receptor desensitization or downregulation. In the same tumors, the treatment with pasireotide, octreotide, or cabergoline at the same concentration of ITF2984 (100 nM) was only effective in a small subset of tumors. Our data are in line with the literature, since in primary cell cultures of NF-PitNETs, it has been shown that octreotide and pasireotide tended to moderately decrease cell viability in a small percentage of tumors and even increased cell viability in another subset of tumors [26,27]. The SSTR2-selective analog BIM23120 was not able to reduce cell proliferation, nor to increase cell apoptosis in NF-PitNET-cultured cells [28], whereas DRD2 agonists induced an in vitro reduction in cell proliferation in about a third of NF-PitNETs [29] and increased cell apoptosis in a subset of tumors [28]. 

Another SSTR3 peptide agonist was previously identified and tested in NF-PitNET primary cultured cells, but it was only effective in a subgroup of tumors, characterized by a higher expression of SSTR3 [13]. 

The analysis of lineage derivation of the NF-PitNETs by RT-qPCR for *SF1* showed that the majority of tumors derived from the gonadotroph cell lineage, as expected. We found no difference in responsiveness to ITF2984 or in SSTR3 expression between *SF1*-positive and -negative tumors. This suggest that a possible clinical application of this drug does not require a stratification of patients with NF-PitNETs based on the tumor type. 

Quantification of the *SSTR3*, *SSTR2*, and *SSTR5* transcript levels in tumor tissues revealed a similar expression of *SSTR3* compared to *SSTR2* and *SSTR5* in our cohort of patients. Other studies analyzing the *SSTR3* transcript in NF-PitNETs have demonstrated both an increased *SSTR3* expression vs. *SSTR2* and *SSTR5* [15,18], and a comparable expression of the three receptors [17]. Interestingly, SSTR3 protein expression correlated with SSTR5, suggesting a possible cooperation of the two receptors in signal transduction. 

We then tested a possible correlation between the expression levels of SSTR3, both at the transcript and at the protein level, and the entity of response to ITF2984. Our data showed no correlation, suggesting that the downstream effect on cell proliferation and apoptosis is not only related to the amount of total receptor expressed in tumoral cells, but also to other factors, including receptor availability at the plasma membrane and an intracellular efficient signal transduction machinery, as previously observed for SSTR2 and SSTR5 [7].

Finally, since a sex-related difference in the expression of SSTR3 and ITF2984 responsiveness has been previously shown in MENX rats [22,30], we analyzed a possible difference in both SSTR3 expression and in vitro responsiveness to ITF2984 in tumors deriving from male vs. female patients. No difference was found in SSTR3 expression or in the responsiveness to ITF2984 in the two groups. The only previous study investigating this aspect only showed a non-significant trend for higher SSTR3 expression in NF-PitNETs removed from female patients [30]. Thus, our data do not support a sex-related patient stratification for treatment with ITF2984, accordingly with the observation that, in most clinical studies, gender was not a predictor of response to treatment with SSTR-targeted therapies [31].

## 4. Materials and Methods

### 4.1. Cell Cultures

The local Ethics Committee previously approved the study, and each patient involved gave informed consent for the use of his/her tumor sample. Human pituitary cells were obtained by transsphenoidal surgery from 23 patients with NF-PitNET. Tissues were partially frozen at −80 °C for subsequent molecular analysis and partially enzymatically dissociated in Dulbecco’s modified essential medium (DMEM) (Gibco, Invitrogen, Life Technologies Inc., Carlsbad, CA, USA) containing 2 mg/mL collagenase (Sigma–Aldrich, St. Louis, MI, USA) at 37 °C for 2 h. Pituitary cells collected by centrifugation were cultured in DMEM containing 10% fetal bovine serum (FBS) (Gibco, Invitrogen, Life Technologies Inc., Carlsbad, CA, USA) and antibiotics. Due to the limited availability of cells deriving from NF-PitNET, the effect of pasireotide, octreotide, and cabergoline was only tested in a subset of primary cultures. The NF-PitNET primary cultures were checked daily by visual inspection with an optical microscope to exclude fibroblast contamination.

### 4.2. Proliferation Assay

Cell proliferation was assessed by the colorimetric measurement of 5-bromo-2-deoxyuridine (BrdU) incorporation during DNA synthesis in proliferating cells according to the instruction of the manufacturer. Briefly, primary cultured cells (3 × 10^4^ cells/well) were seeded in polylysine-coated 96-well plates in a starved medium and incubated for 72 h at 37 °C with increasing concentrations (1 Nm–1000 nM) of ITF2984 (kindly provided by Italfarmaco, Milano, Italy), 100 nM octreotide (Novartis, Basel, Switzerland), 100 nM pasireotide (Recordati Rare Diseases Srl, Milano, Italy), or 100 nM cabergoline (Sigma-Aldrich, St. Louis, MO, USA) and then with BrdU at 37 °C for 24 h. A non-specific binding control, in which cells are incubated without BrdU, was added in each experiment. The assay was performed once for each primary culture tested and each determination was carried out at least in triplicate.

### 4.3. Apoptosis Assay

Caspase-3/7 enzymatic activity was measured using Apo-ONE Homogenous Caspase-3/7 Assay (Promega, WI, USA) according to the instructions of the manufacturer. Primary cultured cells (3 × 10^4^ cells/well) were seeded in polylysine-coated 96-well plates in complete medium and incubated for 48 h at 37 °C with ITF2984 or with octreotide, pasireotide, and cabergoline. 

### 4.4. Real-Time Quantitative PCR (RT-qPCR) Analysis

The total RNA extracted from the frozen tissue was reverse-transcribed (RevertAid H Minus First Strand cDNA Synthesis Kit, Thermo Fisher Scientific, Waltham, MA, USA), and the cDNA was amplified with specific primers for *SF1* and *PIT1* for qualitative RT-PCR (primers available upon request). RT-qPCR was carried out with SsoFastTM EvaGreen^®^ Supermix (Bio-Rad Laboratories, Hercules, CA, USA) in a QuantStudio™ 3 Real-Time PCR System (Applied Biosystems, Thermo Fisher Scientific, Waltham, MA, USA). The primers used for the *SSTR3*, *SSTR2*, and *SSTR5* are available upon request. Data were analyzed with QuantStudioTM Design & Analysis Software v1.5.1, using the ΔCt method.

### 4.5. Western Blot Analysis

In total, 30 µg of total proteins extracted from NF-PitNET tissue samples were analyzed. The primary antibodies used were anti-SSTR3 (1:1000) (Abcam, Cambridge, UK), SSTR2 (1:200) (Santa Cruz Biotechnology, Dallas, TX, USA) and SSTR5 (1:1000) (Thermo-Fisher, Waltham, MA, USA). The secondary antibodies used were anti-rabbit and anti-mouse (1:2000) (Cell Signaling, Denvers, MA, USA). Anti-GAPDH was used as housekeeping (1:4000) (Ambion, Life Technologies, Thermo-Fisher, Carlsbad, CA, USA). Chemiluminescence was detected with a ChemiDOC-IT Imaging System (UVP, Upland, CA, USA) and analyzed with NIH ImageJ 1.52t software.

### 4.6. Statistical Analysis

Variables were reported as median and interquartile range (IQR) along the text and in all graphs. To assess the significance between two series of data, the non-parametric U of Mann–Whitney was applied. Moreover, significance between more than two groups of data was evaluated with the non-parametric Kruskal–Wallis one-way ANOVA test with Dunn’s post hoc test. Correlation was performed by Spearman’s rank correlation test. Data were analyzed with GraphPad Prism 10.1.2 software (GraphPad Software, Inc., La Jolla, CA, USA). *p* < 0.05 was accepted as statistically significant.

## 5. Conclusions

In conclusion, the present study demonstrates an antiproliferative and proapoptotic in vitro effect of ITF2984 in human NF-PitNET cells. Although an inhibitory effect on the growth of spontaneous pituitary tumors in MENX rats has been previously demonstrated [22], it is worth noting that, in the same in vivo model, octreotide and pasireotide exerted significant growth suppression and occasional tumor shrinkage, with superior efficacy for pasireotide [30]. However, clinical data in patients with NF-PitNETs do not confirm these findings, indicating that the MENX rat does not fully recapitulate human NF-PitNETs. In this context, our data in human primary cultured cells deriving from NF-PitNETs are of considerable importance in the evaluation of the potential clinical efficacy of ITF2984. However, further studies testing ITF2984 in other in vivo models, e.g., athymic nude mice xenotransplanted with human NF-PitNETs, are urgently needed to validate ITF2984 as a novel potential pharmacological treatment for this type of tumor. Notably, ITF2984 has completed all preclinical safety studies and it was tested in two phase I clinical trials in normal healthy volunteers and in a phase II clinical trial in acromegalic patients [32,33], facilitating future clinical testing in NF-PitNET patients.

## Figures and Tables

**Figure 1 ijms-25-03606-f001:**
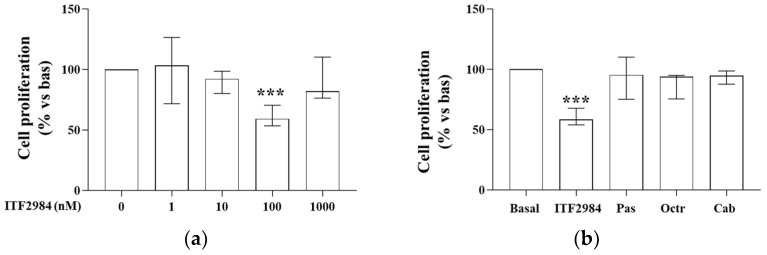
ITF2984 effects on cell proliferation in primary cultured non-functioning pituitary neuroendocrine (NF-PitNET) cells. NF-PitNET cells were incubated for 72 h at 37 °C with increasing concentrations of ITF2984 (**a**) or with 100 nM ITF2984 (*n* = 8), 100 nM pasireotide (*n* = 6), 100 nM octreotide (*n* = 5), and 100 nM cabergoline (*n* = 8) (**b**). BrdU incorporation in newly synthesized DNA was measured. Each determination was carried out in triplicate. The graphs show the median (interquartile range (IQR)). *** *p* < 0.001 vs. corresponding basal. Pas = pasireotide, Octr = octreotide, Cab = cabergoline.

**Figure 2 ijms-25-03606-f002:**
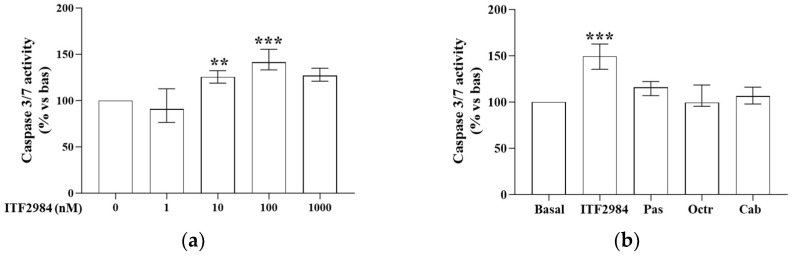
ITF2984 effects on cell apoptosis in primary cultured NF-PitNET cells. NF-PitNET cells were incubated for 48 h at 37 °C with increasing concentrations of ITF2984 (**a**), or with 100 nM ITF2984 (*n* = 10), 100 nM pasireotide (*n* = 7), 100 nM octreotide (*n* = 7), and 100 nM cabergoline (*n* = 10) (**b**). Caspase 3/7 activity was measured. Each determination was carried out in triplicate. The graphs show the median (IQR). ** *p* < 0.01; *** *p* < 0.001 vs. corresponding basal. Pas = pasireotide, Octr = octreotide, Cab = cabergoline.

**Figure 3 ijms-25-03606-f003:**
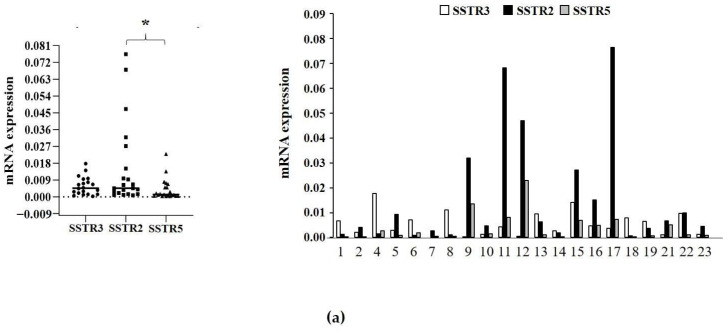
(**a**) Graphs show the expression levels of somatostatin receptor (*SSTR*)-type 3, 2, and 5 transcripts measured by RT-qPCR in NF-PitNET tissues. In the upper graph, medians are shown (horizontal bar). The lower graph shows the expression levels of the three receptors in each NF-PitNET sample. 1–23 = NF-PitNET samples. * *p* < 0.05. (**b**) Protein expression of SSTR3, SSTR2, and SSTR5 normalized to GAPDH and to internal positive control. Proteins extracted from each tissue sample were analyzed by Western blotting and quantified by densitometric analysis. 2–23 = NF-PitNET samples. (**c**) Graph shows Spearman correlation between mRNA levels of *SSTR3* and *SSTR2* or *SSTR3* and *SSTR5*.

## Data Availability

Data available in a publicly accessible repository.

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
