# Peer review of "The Novel SSTR3 Agonist ITF2984 Exerts Antimitotic and Proapoptotic Effects in Human Non-Functioning Pituitary Neuroendocrine Tumor (NF-PitNET) Cells"

_ijms, 2024, doi:10.3390/ijms25073606_

Round 1
Reviewer 1 Report
Comments and Suggestions for Authors
The authors demonstrate an antimitotic and proapoptotic effect of the novel SSTR3 agonist ITF2984 in human non-functioning pituitary neuroendocrine tumor (NF-PitNET) in cell culture (Actually, ITF2984 is described as a pan-SSTR agonist with high affinity for SSTR3).
The biggest antiproliferative effect on NF-PitNET cells is found at a concentration of 100 nM. Importantly, no significant effect is found at this concentration with pasireotide, octreotide or cabergoline. Similiarly, a significant increase of apoptosis was demonstrated with ITF2984 but not with pasireotide, octreotide or cabergoline. It is an innovative and important study that shows a possible use of the novel SSTR3 agonist in the treatment of NF-PitNETs. It is the first study investigating the effect of ITF2984 on proliferation of human NF-PitNET cells.
The study underlines the superiority of studies with human adenoma cells when compared to available animal models.
The authors should explain under Methods the rationale for the different number of cell cultures which have been used for the different drugs (f.e. proliferation experiments ITF2984 N=19, pasireotide N=6, octreotide N=5, cabergoline N=8). The sentence in the Abstract „We treated cells derived from 23 NF-PitNETs with ITF2984, octreotide, pasireotide or cabergoline …“ is misleading as it makes believe that a high number of cell cultures were used for all drugs.
Author Response
REVIEWER 1
We thank the reviewer for the careful reading of our manuscript and for her/his comments and suggestions. The topics raised by the reviewer have been all addressed. In particular:
The authors should explain under Methods the rationale for the different number of cell cultures which have been used for the different drugs (f.e. proliferation experiments ITF2984 N=19, pasireotide N=6, octreotide N=5, cabergoline N=8).
We thank the reviewer for the suggestion. We added a sentence in the Material and Methods section explaining that the reason why we used different number of cell cultures to test different drugs was due to the limited availability of cells derived from human NF-PitNET (Materials and Methods section, lines 246-248). In particular, all proliferation experiments were performed with the drug under study (ITF2984), whereas only a subset of tumors has also been incubated with the other drugs.
The sentence in the Abstract „We treated cells derived from 23 NF-PitNETs with ITF2984, octreotide, pasireotide or cabergoline …“ is misleading as it makes believe that a high number of cell cultures were used for all drugs.
We thank the reviewer for the comment. The sentence in the abstract has been edited to make it clearer (Abstract section, line 40).

Reviewer 2 Report
Comments and Suggestions for Authors
The article titled "The novel SSTR3 agonist ITF2984 exerts antimitotic and proapoptotic effects in human non-functioning pituitary neuroendocrine tumor (NF-PitNET) cells" aims to test the antiproliferative effects of ITF2984 on non-functioning pituitary neuroendocrine tumor (NF-PitNET) cells. The article is of good quality and answers the proposed question. However, there is still space to improve the article.
The main point I make is about the figures in the article. The authors include all the graphs, but they lack representative figures of the results described. Western blot markings and results must be accompanied by graphs showing the results.
Author Response
REVIEWER 2
We thank the reviewer for the careful reading of our manuscript and for her/his comment.
The main point I make is about the figures in the article. The authors include all the graphs, but they lack representative figures of the results described. Western blot markings and results must be accompanied by graphs showing the results.
We thank the reviewer for the comment. As suggested, we added in the Figure 3b representative Western blots related to the graphs (Revised Figure 3).
